# Obstructions and the Recognition of Cancer Inpatient Physical Activity Based on Exercise Experience

**DOI:** 10.3390/ijerph17155482

**Published:** 2020-07-29

**Authors:** Jeon Sangwan, Yi Eunsurk, Kim Jiyoun

**Affiliations:** 1Exercise Rehabilitation Convergence Institute, Gachon University191 Hombakmoero, Yeonsu-gu, Incheon 406-799, Korea; jsw3972@kspo.or.kr; 2Department of Exercise Rehabilitation and Welfare, Gachon University191 Hombakmoero, Yeonsu-gu, Incheon 406-799, Korea; yies@gachon.ac.kr

**Keywords:** physical activity constraints, cancer, inpatients, exercise experience, exercise cognition

## Abstract

The purpose of this study was to analyze and understand the mechanisms of physical activity obstructions in hospitalized cancer patients by investigating their physical activity levels, previous exercise experience levels, and exercise recognition. A survey was conducted for 194 hospitalized cancer patients using a questionnaire. In addition, we performed exploratory factor analysis, frequency analysis, reliability analysis, and hierarchical multiple regression analysis, using SPSS Statistics for Windows, Ver. 23.0. The results were as follows: (1) The physical activity level of the previous exercise participation experience (EPE) group had a greater effect on physical activity obstructions compared with the non-experience (NE) group. (2) The results for the effects of exercise recognition on the physical activity level and physical activity obstructions indicated that exercise recognition in the two groups increased the relative effects on physical activity obstructions in all variables except for the physical obstructions of the EPE group. Consequently, the physical activity level, exercise experience level, and exercise recognition in those patients were confirmed to be the major factors affecting their physical activity obstruction. Therefore, in this study, we provided quantitative data required for establishing healing environments based on motion.

## 1. Introduction

Cancer accounts for a significant portion of deaths around the world [1]. In Korea, more people die of cancer than of any other disease. These results are attributed to a 10% annual increase in the number of cancer patients by aging, westernized eating habits, and altered lifestyles [2].

According to the Basic Psychological Need Theory [3], the psychological desire of satisfaction consists of autonomy, ability, and relationships, which explains it as an important prior variable that impacts behavior, emotions, cognition, and motivation levels. In particular, negative experiences of past exercise not only provide physical and mental fatigue [4] but result in a physically inactive sedentary life [5]. In addition, the experience of participating in sports at an early age is the basis for the formation of lifelong exercise habits [6], which not only confirms exercise as a means of maintaining good health, but also helps to pursue a life that goes beyond external decline, such as aging and disease [7].

The majority of patients with chronic diseases are negatively affected by physical activity due to their symptoms [8]. Participation in cancer-related physical activity has a positive effect on their physical image, self-esteem, emotional well-being, sleep disorders, social functions, anxiety, and fatigue [9]. In addition, the recovery of functions through physical activity is closely related to the persistence of physical activity [10]. Therefore, the experience of physical activity is likely to affect the participation in physical activities within the hospital admission period.

Physical activity is currently considered valuable. However, despite its effects, there are many obstructions for physical activities for hospitalized patients owing to the spatial obstructions of hospitals [11]. Therefore, these obstructions constrain the patient’s daily life and significantly reduce physical functions and activities when in the hospital [12]. Physical activity and exercise are also excluded for other reasons, even though they consist of medical services for treatment and recovery. Yang et al. [13] reported that medical experts acknowledged the positive effects of physical exercise rehabilitation in improving many cancer patients, but actually failed to implement them due to spatial obstructions.

Previous studies have been conducted in cancer patients to present exercises (including aerobic, muscle, and complex exercise) and verify the effects through exercise programs [14], as well as to study what is required for treatment, recovery, and prevention [15,16]. Therefore, based on prior research that showed how physical activity is effective for curing and recovering from cancer, we intended to compare and analyze the levels of physical activity and the influence of exercise recognition and exercise obstructions according to a patient’s previous exercise experience (EPE group: exercise participation experience; NE group: non-experience).

Through this, we provided an important basis of data to inform proper physical activity habits for cancer recovery and healing activities and basic data for systematic information provisions. We also proved that the foundation for recovery, health promotion, and maintenance after clinical treatment was in human movement. Moreover, we presented an opportunity to enhance the value of new medical and health services by providing a system to strengthen them, and establish a patient-centered health and medical service environment to use as a basis for public relations and education regarding cancer management at the preventive and recovery level.

## 2. Methods

### 2.1. Subjects

This study selected adults aged 20 years or more residing in metropolitan areas (such as Seoul, Gyeonggi, and Incheon) as a population, and then 250 cancer patients as subjects using purposive sampling from five general hospitals located in the selected areas. The specific criteria for the selection included adults over the age of 20 who were diagnosed with cancer and hospitalized in a specialized hospital for treatment. Among them were 8 patients under 40 years of age (EPE 4, NE 4), 32 patients in their 40 s (EPE 16, NE 16), 69 patients in their 50 s (EPE 34, NE 35), 48 patients in their 60 s (EPE 24, NE 24), and 37 patients over 70 years old (EPE 19, NE 18). In addition, people who exercised more than three days a week and more than 30 min a day for the past year before their diagnosis of cancer were classified as having physical activity experience. After explaining the purpose of the study, only those who signed the consent form participated in the study. A survey was performed on each subject from March to May 2017. We excluded 56 collected questionnaires due to omitted or faithless answers. Lastly, we employed 194 questionnaires in this study, as shown in Figure 1.

### 2.2. Research Tools

We applied a questionnaire to analyze the mechanism of physical activity experience and obstructions in hospitalized cancer patients in Korea. A questionnaire for exercise recognition was modified and supplemented to serve the study purpose based on previously proposed questionnaire for exercise attitudes [17,18], and for exercise decisional balance [19,20]. A questionnaire for the cause of physical activity obstruction was also modified and supplemented to fit the contents and subjects based on the questionnaire used [21,22]. After a preliminary survey in 60 cancer patients, we eliminated items with a factor loading of less than 0.5, and constructed a questionnaire with a total of 42 items, specifically including 5 items for background variables, 6 items for medical history, 10 for exercise recognition, 8 for environmental obstruction, and 13 for personal obstruction. To help subjects understand the contents of the questionnaire, we modified and supplemented the contents using simple terms based on the review of five professors in the departments of nursing, physical therapy, and exercise rehabilitation.

### 2.3. Reliability and Validity of the Questionnaire

Prior to the survey, we requested reviews and comments for the questionnaire in an expert meeting composed of doctors in nursing, physical therapy, exercise rehabilitation, medicine, sports sociology, and exercise physiology, and subsequently discussed the content validity and item relevancy of the questionnaire. The final validity was verified using confirmatory factor analysis on the basis of collected materials. Principal component analysis was first conducted to extract the constituent factors. Subsequently, we selected Varimax as an orthogonal rotation method to simplify the factor loadings and used the Kaiser–Meyer–Olkin (KMO) measure as a reference value to select variables for factor analysis. Bartlett’s test of sphericity (χ^2^) was also employed to verify the suitability of the factor analysis and discover common factors. For selection standards, only factors with eigenvalues of 1.0 or more during extraction were accepted.

For exercise recognition, confirmatory factor analysis showed a KMO of 0.890, a χ^2^ of 1083.397, an eigenvalue of 5.044 in positive exercise recognition, and an eigenvalue of 1.170 in negative exercise recognition. For physical activity obstruction, we found a KMO of 0.625, a χ^2^ of 973.351, an eigenvalue of 2.376 in physical obstruction, an eigenvalue of 2.179 in cognitive psychological obstruction, an eigenvalue of 2.017 in socio-cultural obstruction, an eigenvalue of 2.206 in facility obstruction, and an eigenvalue of 1.790 in program participation obstruction. Cronbach’s coefficient alpha was used for the reliability analysis. Factor loadings of 0.5 or more were all accepted as a reference to select questionnaire items. Consequently, five items in the part of physical activity obstruction were removed. All other factors obtained the validity and reliability coefficient (positive exercise recognition = 0.927, negative exercise recognition = 0.761, facility obstruction = 0.657, program obstruction = 0.709, cognitive psychological obstruction = 0.783, socio-cultural obstruction = 0.731, and physical obstruction = 0.675).

### 2.4. Data Analysis

This study tried to analyze the mechanism of physical activity experience and obstruction in cancer patients during hospitalization. For the survey, we asked for cooperation from hospital officials and then investigators and sub-investigators directly visited the relevant hospitals. They sufficiently explained purposes of the survey and how to fill out the questionnaire to the subjects. This survey employed a self-administered questionnaire. In accordance with the data collecting method of this study, we excluded a number of questionnaires due to faithless, double, and omitted answers. We coded the individual data we believed to be reliable into a computer. They were processed through SPSS Statistics for Windows, Ver. 23.0 (IBM Co., Armonk, NY, USA) to serve the study purpose. In addition, statistical methods, such as frequency analysis, exploratory factor analysis, reliability analysis, and hierarchical multiple regression analysis, were introduced in this study.

## 3. Results

We divided the levels of physical activity. The hours of physical activity for sedentary life, daily life, and recovery were divided into the hours of activities for daily activities, such as washing, going to the bathroom, etc., and the hours of physical activity were divided into the hours for artificially promoting physical activities for recovery.

Prior to multiple regression analysis to investigate the effects of physical activity levels on physical activity obstructions in hospitalized cancer patients, we conducted residual analysis for the regression model and found that there were no outliers. The model did not violate the normality, homoscedasticity, or linearity of the residual. In the regression model for the physical activity level and physical activity obstructions, the F values were found to be statistically significant as shown in Table 1. Thus, the model was considered appropriate based on its linearity.

Multiple regression analysis for the effect of the physical activity level on physical activity obstructions in the EPE and NE groups revealed the following results, as summarized in Table 1 and Figure 2. First, the physical activity level of the EPE group partially affected facility obstructions. In terms of the relative importance of the influence, this had a greater effect on the physical activity time (β = 0.463) compared to the sedentary time (β = 0.394). The explanation power of the effect of the physical activity level on facility obstructions was 30.5% in the EPE group. Similarly, in the NE group, the physical activity level partially affected the facility obstructions. In terms of the relative importance of the influence, the physical activity time (β = 0.254) had a greater effect on the facility obstructions than did the daily life time (β = −0.215). The explanation power for the effect of the physical activity level on facility obstructions was 6.1%.

Second, in the EPE group, the physical activity level had a partial effect on the program obstructions. In addition, the physical activity time (β = 0.424) and the daily life time (β = −0.351) had different effects depending on the importance. The explanation power for the effect of the physical activity level on program obstructions was 24.2%.

Third, the physical activity level in the EPE group affected only the sedentary time (β = 0.617), which is an independent variable of physical obstructions. The explanatory power for the effect of the physical activity level on physical obstructions was 33.4%.

Fourth, in the EPE group, the physical activity level partially affected the cognitive psychological obstructions. Specifically, the cognitive psychological obstructions were affected by two independent variables differently depending on importance: sedentary time (β = 0.359) and physical activity time (β = −0.381). The explanatory power for the effect of the physical activity level on cognitive psychological obstructions was 30.8%. However, the physical activity level in the NE group only affected the sedentary time (β = 0.255), which is an independent variable of cognitive psychological obstructions.

Fifth, the physical activity level in the EPE group only affected the sedentary time (β = 0.343), which is an independent variable of socio-cultural obstructions. The explanatory power for this effect was 10.9%.

Prior to the hierarchical regression analysis to investigate the effects of the physical activity levels and exercise recognition on physical activity obstructions, we conducted a residual analysis of the regression model and found that there were no outliers. In addition, the model did not violate the normality, homoscedasticity, and linearity of the residual. In the regression model for the physical activity level, exercise recognition, and physical activity obstruction, the F values were found to be statistically significant as shown in Table 2. Thus, the model was considered appropriate based on its linearity.

Hierarchical regression analysis for the effect of physical activity level and exercise recognition on physical activity obstructions in EPE and NE groups produced the following results, as summarized in Table 2 and Figure 2, Figure 3 and Figure 4. First, for the EPE group, Model 2 had a greater effect on the level of physical activity and exercise recognition compared to Model 1. The addition of exercise recognition increased the relative influence on facility obstructions from 30.5% to 31.3%, and physical activity time (β = 0.453) was also shown to have a significant statistically significant effect. Similarly, in the NE group, the physical activity level and exercise recognition had a greater effect on the facility obstructions in Model 2 compared to Model 1. Added exercise recognition affected the facility obstructions (13.9%). This had a significant positive effect on the positive exercise recognition and physical activity time (*p* < 0.01, *p* < 0.05), and a significant negative effect on the daily life time (*p* < 0.01). In particular, it had the largest effect on positive exercise recognition (β = 0.311), followed by daily life time (β = −0.263) and physical activity time (β = 0.218).

Second, in the EPE group, the physical activity level and exercise recognition had a greater effect on program obstructions in Model 2 compared to Model 1Added exercise recognition had an impact on the program obstructions (44.6%). This exerted significant positive influence on the physical activity time and negative exercise recognition (*p* < 0.001), and a significant negative influence on the daily life time and positive exercise recognition (*p* < 0.001, *p* < 0.05). In particular, it had the largest effect on physical activity time (β = 0.496), followed by negative exercise recognition (β = 0.414), daily life time (β = −0.383), and positive exercise recognition (β = −0.198). Similarly, in the NE group, the physical activity level and exercise recognition had a greater effect on the program obstructions in Model 2 compared to Model 1. Added exercise recognition affected the program obstructions (6.2%) and had a significant positive effect on negative exercise recognition (*p* < 0.01).

Third, in the EPE group, the physical activity level and exercise recognition had an effect on the physical obstructions equally in Model 1 and Model 2. Added exercise recognition affected the physical obstructions (33.4%) and had a significant positive effect on the sedentary time (*p* < 0.001). However, in the NE group, the physical activity level and exercise recognition had a greater effect on the physical obstructions in Model 2 compared to Model 1. Added exercise recognition affected the physical obstructions (22.0%). This had significant positive and negative effects on negative exercise recognition and positive exercise recognition, respectively (*p* < 0.05, *p* < 0.001). In terms of the relative influence, positive exercise recognition (β = −4.672) was affected the most, followed by negative exercise recognition (β = 2.870).

Fourth, in the EPE group, the physical activity level and exercise recognition had a greater effect on the cognitive psychological obstructions in Model 2 compared to Model 1. Added exercise recognition affected the cognitive psychological obstructions (65.0%). This had a significant positive effect on negative exercise recognition, where the sedentary time had a significant negative effect on positive exercise recognition and the physical activity time (*p* < 0.001). In addition, it exerted the greatest influence on negative exercise recognition (β = 0.449), followed by sedentary time (β = 0.342), positive exercise recognition (β = −0.371), and physical activity time (β = −0.270). Similarly, in the NE group, the physical activity level and exercise recognition had a greater effect on cognitive psychological obstructions in Model 2 compared to Model 1. Added exercise recognition affected the cognitive psychological obstructions (68.4%). This had a significant positive effect on the daily life time, sedentary time, and negative exercise recognition (*p* < 0.05, *p* < 0.01, *p* < 0.001), and a significant negative effect on positive exercise recognition (*p* < 0.001). In addition, for the relative influence, negative exercise recognition (β = 0.692) was affected the most, followed by positive exercise recognition (β = −0.370), sedentary time (β = 0.185), and daily life time (β = 0.141).

Fifth, in the EPE group, the physical activity level and exercise recognition had a greater effect on the socio-cultural obstructions in Model 2 compared to Model 1. Added exercise recognition affected the socio-cultural obstructions (14.2%). This had positive and negative effects on the sedentary time and positive exercise recognition, respectively (*p* < 0.01 *p* < 0.05). In addition, the former (β = 0.364) was more affected than the latter (β = −0.236).

## 4. Discussion

The present study was conducted to empirically investigate the mechanism of physical activity experience and obstructions in hospitalized domestic cancer patients. Based on this, we classified patients based on their exercise experience and analyzed how the physical activity level and exercise recognition were correlated to, and different from, physical activity obstructions. In addition, we classified the EPE group separately and explored the effect of the exercise experience level on physical activity obstructions. Here, we discuss the results as follows.

First, regarding the physical activity level and physical activity obstructions in hospitalized domestic cancer patients, our analysis of the effects of exercise recognition showed that exercise recognition affected facility obstructions in the EPE and NE groups. In particular, in the NE group with higher influence, the subjects recognized the necessity of exercise at a lower level in the past. However, after cancer development, they began to recognize the need for physical activities to improve and treat cancers. Naturally, their needs and interests in exercise facilities may have a great effect on facility obstructions.

On the other hand, subjects in the EPE group may be good at adapting to and using exercise facilities, as they understand exercise and its methods at a higher level. In the end, patient-oriented medical services must focus on patients as subjects for treatment and respect their demands, rights, and interests. Currently, contemporary medicine does not translate narrow health concepts from a physical viewpoint and contends that hospital construction should be altered to be a place for recovering patients beyond the old concepts of function, standardization, and rationalization. Other research mentions the importance of healing environments at the same time [23] and supports our study results.

For program obstruction, exercise recognition in the EPE group exerted a higher influence on program obstructions compared to the NE group. In particular, in the EPE group, positive and negative exercise recognition affected program obstructions, conflicting with each other. This suggests that positive exercise experience in the past affects the obstructions in the process of self-rationalizing of even poor exercise programs of the hospital by recognizing the effect and need for exercise. Negative exercise recognition affects the obstructions owing to the less-than-expected exercise programs for cancers. In this regard, previous works [24,25] have reported that patients were reluctant to take part in an exercise program if the program was contrary to their expectations. Generally, patients with cancers or other serious diseases have high expectations of diverse interventional treatments, including exercise, nutrition, and supplements, due to their strong desire to extend and recover life. Thus, unless the program meets their expectations, they hesitate to participate in the program, with disappointment. In particular, a lack of exercise information may act as a participation obstruction factor in cancer patients undergoing chemotherapy [26]. This information is now provided from hospitals at the “modest” level. Practically, there were many obstructions [27].

In the EPE group, exercise recognition did not have any influence on physical obstructions. Conversely, it greatly affected physical obstructions in the NE group. We considered that patients in the NE group demanded the positive needs of exercise as a measure to improve their health, with a dramatic change in exercise recognition after cancer development. Contrary to this, negative exercise recognition had an effect on physical obstructions due to the presence of negative recognition for exercise. In particular, we expected physical obstructions in cancer patients, as their physical activities lead more strongly to injury, worsened health, and pain after cancer development [28].

For cognitive psychological obstructions, exercise recognition had a huge effect in the EPE and NE groups. This suggests that personal obstructions acted more importantly as an influencing factor for exercise recognition, compared to environmental obstructions, such as facility and program obstruction. According to a previous study [29], cancer patients have a high level of cognitive and empirical stress during the early stages of exercise, so recognizing the merits of exercise through evaluation and exercise counseling by doctors and managers affects the vitalization of physical activity.

Through this process, physical activities and associated skills, such as goal setting and problem solving for disability, affected social support and self-efficacy. This suggests that, consistent with prior studies, patients suffering from cancers accept positive physical activities that can be effective for recovery and recognize the value of exercise through physical activities by themselves despite poor environments, rationalizing this situation in order to feel less obstructions. In addition, for negative exercise recognition in the EPE group, subjects in the group felt a sense of difference for the new environment (that is, the hospital) differing from their existing exercise environment, despite having exercise experience in the past. For instance, subjects may be conscious of others or establish a negative recognition for exercise owing to a resistance to the new exercise methods proposed by doctors.

On the other hand, in the NE group, exercise recognition had positive and negative effects on cognitive psychological obstructions, as the doctors and those around the patients put emphasis on the importance of exercise to the patients, who considered the exercise as important. In this regard, prior studies indicated that increased recognition for health care led to physical activities higher than those of middle-aged adults by virtue of cancer diagnosis [30]. In addition, evidence for the effects of physical activities on cancer have rapidly increased. These results suggest that physical activities of hospitalized cancer patients are now highlighted as a nonpharmacological intervention [31].

Lastly, positive exercise recognition did not affect socio-cultural obstructions in either group. Generally, forcing cancer patients to perform exercise by themselves can be another cause of stress. Accordingly, it is important to create environments so that they are able to exercise [32]. Family, medical teams, and exercise professionals should consistently motivate them to have a positive impression of exercise. This will contribute to reducing socio-cultural obstructions.

A number of prior studies revealed that exercise can have a very beneficial effect on cancer patients. Aerobic exercise, in particular, induces the contraction and relaxation of blood vessels while contracting and relaxing the muscles. This can enhance blood vessel function, increase blood flow to the heart, strengthen heart function [33], and improve the resistance to muscular fatigue by preventing leukocyte reduction [34]. This results in an average maximum oxygen intake of 14 mL/kg/min for cancer patients, which is approximately 25% of non-cancer patients, reaching a level where even a small amount of physical activity can cause severe fatigue [35]. However, after the application of motor therapy, the maximum oxygen intake was shown to increase significantly, which is also positive for the prognosis of patients, and has the effect of improving the musculoskeletal system and mitigating the side effects of chemotherapy [36]. This is also effective for the purpose of preventing or treating psychological problems as well as for the recovery of these functions [37].

Therefore, in order to aid in hospitalized cancer patient recovery, the physical activity experience of cancer patients and the mechanism of disability should be carefully analyzed so that appropriate responses to exercise at the health and medical service level can be attempted.

## 5. Conclusions 

This study was designed to analyze the obstruction factors that inpatients experience when participating in physical activities. We focusing on personal and environmental characteristics, actual obstacles, and how they related to the differences between physical activity levels and exercise recognition. Through this, we attempted to provide the importance of physical activities and environment for the recovery of health and the improvement of the quality of life as basic data for the improvement of health arbitration by identifying factors affecting the physical activities of inpatients.

According to this study, the differences between physical activity levels, exercise recognition, and physical activity obstructions based on the pre-admission experiences of cancer patients were found to be statistically significant in terms of cognitive psychology, social culture, and facility obstructions. Second, the results of the hierarchical regression analysis showed that the level of physical activity and the influence of exercise recognition on physical activity obstructions based on exercise participation experience showed that a sedentary life, intentional movement, and negative exercise recognition were the most important factors of physical activity obstruction in the case of the exercise participation experience group. Positive exercise recognition and a negative social culture were the most important factors in the case of the exercise nonparticipating experience group.

That is, in the case of the exercise participation experience groups, physical activity obstructions were found to be directly related to physical activity as this group was more aware of the habits and effectiveness of exercise acquired through past exercise experiences, and their desire for active physical activity was relatively large. On the other hand, in the case of the non-participating groups, we found the physical activity constraints to be highly related to the socio-psychological factors. Both groups commonly perceived exercise as an important factor in physical activity obstructions.

Therefore, we would like to make the following suggestions. Hospitals and health care facilities should make efforts to restore the health of patients in specialized cancer nursing hospitals and to extend their lives and improve their quality of life. To that end, they should consider the environmental construction to promote the physical activities of inpatients, the conversion of exercise recognition, support for medical personnel, measures to improve family support, and strategies to enhance patient-centered satisfaction with their medical services.

## Figures and Tables

**Figure 1 ijerph-17-05482-f001:**
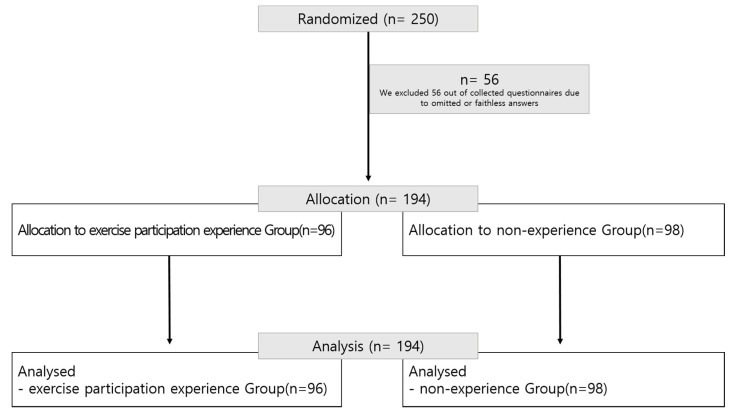
Participant allocation (consolidated standards for reporting the trial as a flow diagram).

**Figure 2 ijerph-17-05482-f002:**
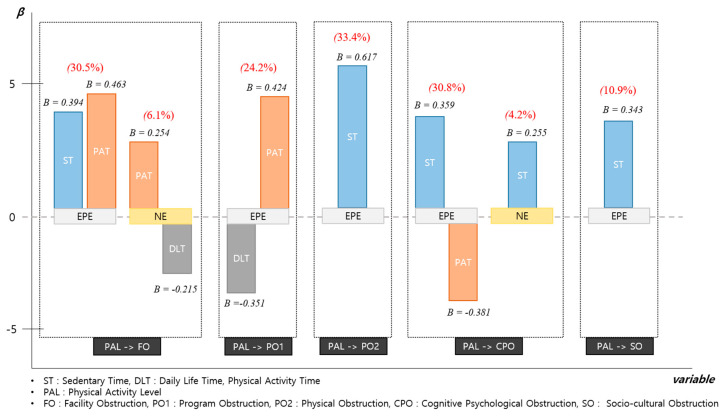
Hierarchical regression analysis for the effects of the physical activity levels and obstruction factors based on the physical activity participation experience.

**Figure 3 ijerph-17-05482-f003:**
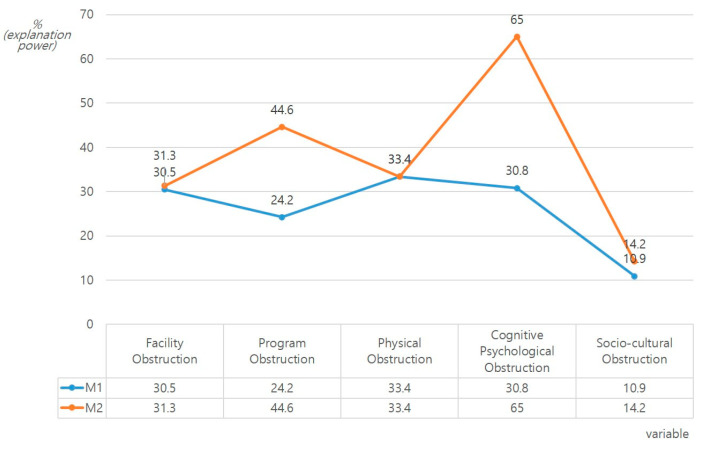
Results of the hierarchical regression analysis for the effects of the physical activity levels and obstruction factors based on the physical activity participation experience (EPE).

**Figure 4 ijerph-17-05482-f004:**
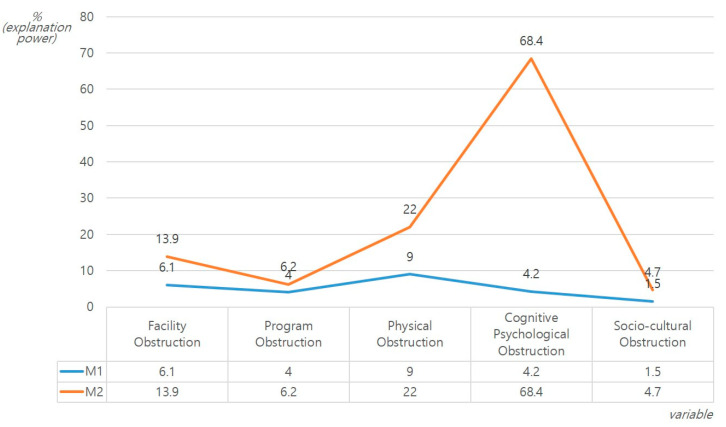
Results of the hierarchical regression analysis for the effects of the physical activity levels and obstruction factors based on the physical activity participation experience (NE).

**Table 1 ijerph-17-05482-t001:** Results of the multiple regression analysis of the physical activity levels and obstruction factors based on the physical activity participation experience. EPE group: exercise participation experience; NE group: non-experience.

Independent Variable	Dependent Variable	EPE Group			NE Group		
		*Β*	*T*	*p*	*Β*	*T*	*p*
Sedentary time	facility obstruction	0.394	4.076	0.000	−0.005	−0.057	0.954
Daily life time		−0.080	−0.821	0.414	−0.215	−2.299	0.023
Physical activity time		0.463	4.901	0.000	0.254	2.709	0.008
*F* = 12.535 ***, *R*^2^ = 0.305, *D* − *W* = 2.028					*F* = 3.442 *, *R*^2^ = 0.061, *D* − *W* = 1.643		
Sedentary time	program obstruction	0.160	1.586	0.117	0.056	0.596	0.552
Daily life time		−0.351	−3.475	0.001	0.000	−0.005	0.996
Physical activity time		0.424	4.302	0.000	0.139	1.433	0.155
*F* = 9.423 ***, *R*^2^ = 0.242, *D* − *W* = 1.695					*F* = 0.840, *R*^2^ = −0.004, *D* − *W* = 1.550		
Sedentary time	physical obstruction	0.617	6.518	0.000	0.144	1.542	0.126
Daily life time		−0.174	−1.834	0.071	−0.118	−1.223	0.224
Physical activity time		0.079	0.859	0.393	−0.010	−0.107	0.915
*F* = 14.212 ***, *R*^2^ = 0.334, *D − W* = 1.903					*F* = 0.1.357, *R*^2^ = 0.009, *D* − *W* = 1.431		
Sedentary time	cognitive psychological obstruction	0.359	3.715	0.000	0.255	2.773	0.007
Daily life time		0.158	1.638	0.106	0.018	0.189	0.850
Physical activity time		−0.381	−4.038	0.000	−0.044	−0.464	0.644
*F* = 12.727 ***, *R*^2^ = 0.308, *D* − *W* = 2.230					*F* = 2.633 *, *R*^2^ = 0.042, *D* − *W* = 1.863		
Sedentary time	socio-cultural obstruction	0.343	3.134	0.002	−0.142	−1.522	0.131
Daily life time		0.080	0.729	0.468	0.110	1.145	0.255
Physical activity time		−0.060	−0.563	0.575	−0.123	−1.282	0.202
*F* = 4.217 ***, *R*^2^ = 0.109, *D* − *W* = 2.486					*F* = 1.835 *, *R*^2^ = 0.015, *D* − *W* = 1.835		

* *p* < 0.05, *** *p* < 0.001.

**Table 2 ijerph-17-05482-t002:** Results of the hierarchical regression analysis for the effects of the physical activity levels and obstruction factors based on the physical activity participation experience.

Model	Independent Variable	Dependent Variable	EPE Group			NE Group		
			*Β*	*T*	*p*	*Β*	*T*	*p*
1	Sedentary time	facility obstruction	0.394	4.076	0.070	−0.005	−0.057	0.954
	Daily life time		−0.080	−0.821	0.060	−0.215	−2.299	0.023
	Physical activity time		0.463	4.901	0.051	0.254	2.709	0.008
*F* = 12.535 ***, *R*^2^ = 0.305, *D* − *W* = 2.030						*F* = 3.442 ***, *R*^2^ = 0.061, *D* − *W* = 1.526		
2	Sedentary time	facility obstruction	0.370	3.797	0.068	0.022	0.247	0.806
	Daily life time		−0.077	−0.793	0.058	−0.263	−2.887	0.005
	Physical activity time		0.453	4.710	0.049	0.218	2.404	0.018
	Positive recognition		0.093	0.969	0.083	0.311	3.463	0.001
	Negative recognition		0.135	1.431	0.069	0.003	0.030	0.976
*F* = 8.191 ***, *R*^2^ = 0.313, *D* − *W* = 2.030						*F* = 4.658 ***, *R*^2^ = 0.139, *D* − *W* = 1.526		
1	Sedentary time	program obstruction	0.160	1.586	0.081	0.056	0.596	0.552
	Daily life time		−0.351	−3.475	0.069	0.000	−0.005	0.996
	Physical activity time		0.424	4.302	0.058	0.139	1.433	0.155
*F* = 9.423 ***, *R*^2^ = 0.242, *D* − *W* = 1.706						*F* = 0.840, *R*^2^ = −0.004, *D* − *W* = 1.338		
2	Sedentary time	program obstruction	0.131	1.497	0.079	0.041	0.448	0.655
	Daily life time		−0.383	−4.413	0.068	0.026	0.273	0.785
	Physical activity time		0.496	5.740	0.057	0.115	1.215	0.227
	Positive recognition		−0.198	−2.305	0.097	0.007	0.077	0.938
	Negative recognition		0.414	4.887	0.080	0.288	3.127	0.002
*F* = 13.720 ***, *R*^2^ = 0.446, *D* − *W* = 1.706						*F* = 2.497 *, *R*^2^ = 0.062, *D* − *W* = 1.338		
1	Sedentary time	physical obstruction	0.617	6.518	0.071	0.144	1.542	0.126
	Daily life time		−0.174	−1.834	0.060	−0.118	−1.223	0.224
	Physical activity time		0.079	0.859	0.051	−0.010	−0.107	0.915
*F* = 14.212 ***, *R*^2^ = 0.334, *D* − *W* = 1.961						*F* = 1.357, *R*^2^ = 0.009, *D* − *W* = 1.577		
2	Sedentary time	physical obstruction	0.637	6.651	0.063	0.096	1.155	0.251
	Daily life time		−0.181	−1.899	0.054	−0.033	−0.385	0.701
	Physical activity time		0.098	1.036	0.046	0.016	0.190	0.850
	Positive recognition		−0.114	−1.214	0.077	−0.399	−4.672	0.000
	Negative recognition		−0.074	−0.792	0.064	0.241	2.870	0.005
*F* = 8.940 ***, *R*^2^ = 0.334, *D* − *W* = 1.961						*F* = 7.390 ***, *R*^2^ = 0.220, *D* − *W* = 1.577		
1	Sedentary time	cognitive psychological obstruction	0.359	3.715	0.064	0.255	2.773	0.007
	Daily life time		0.158	1.638	0.055	0.018	0.189	0.850
	Physical activity time		−0.381	−4.038	0.046	−0.044	−0.464	0.644
*F* = 12.727 ***, *R*^2^ = 0.308, *D* − *W* = 2.432						*F* = 2.633 *, *R*^2^ = 0.042, *D* − *W* = 2.045		
2	Sedentary time	cognitive psychological obstruction	0.342	4.923	0.037	0.185	3.484	0.001
	Daily life time		0.111	1.607	0.032	0.141	2.555	0.012
	Physical activity time		−0.270	−3.930	0.027	−0.057	−1.036	0.303
	Positive recognition		−0.371	−5.424	0.046	−0.370	−6.805	0.000
	Negative recognition		0.449	6.671	0.038	0.692	12.963	0.000
*F* = 30.361 ***, *R*^2^ = 0.650, *D* − *W* = 2.432						*F* = 49.930 ***, *R*^2^ = 0.684, *D* − *W* = 2.045		
1	Sedentary time	socio-cultural obstruction	0.343	3.134	0.065	−0.142	−1.522	0.131
	Daily life time		0.080	0.729	0.056	0.110	1.145	0.255
	Physical activity time		−0.060	−0.563	0.047	−0.123	−1.282	0.202
*F* = 4.217 **, *R*^2^ = 0.109, *D* − *W* = 2.618						*F* = 1.591, *R*^2^ = 0.015, *D* − *W* = 1.827		
2	Sedentary time	socio-cultural obstruction	0.364	3.348	0.065	−0.162	−1.759	0.081
	Daily life time		0.060	0.551	0.055	0.146	1.522	0.131
	Physical activity time		−0.009	−0.087	0.047	−0.099	−1.036	0.303
	Positive recognition		−0.236	−2.201	0.079	−0.223	−2.361	0.020
	Negative recognition		0.018	0.168	0.066	o.017	0.854	0.854
*F* = 3.607 **, *R*^2^ = 0.142, *D* − *W* = 2.618						*F* = 2.123, *R*^2^ = 0.047, *D* − *W* = 1.827		

* *p* < 0.05, ** *p* < 0.01, *** *p* < 0.001.

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
