# Peer review of "Obstructions and the Recognition of Cancer Inpatient Physical Activity Based on Exercise Experience"

_ijerph, 2020, doi:10.3390/ijerph17155482_

Round 1

Reviewer 1 Report

The reader understands that there are hospital architectural limitations to physical activity during cancer treatment. The separation into sports and non-sports groups makes sense. On the other hand, in the absence of the questionnaire it is difficult to follow the logic of the work.

The internet link does not work, it would be advisable to publish it as an annex.

The focus of the research needs to be better explained

Many expressions need to be clarified:

Obstruction: do you mean limitation, disruption...

Recognition: do you mean awareness, knowledge, perception, ability...

l 105: program obstruction

l113: hospitalized domestic cancer patients?

l138: Daily life time ? What is the concept?

l244: patent probably patient

l246: diver probably diverse

The conclusion should better relate the results to the problematic. The practical implications need to be better described.

Author Response

Thank you for your insightful and useful comments.

We specified the research method, corrected the errors in the statistical method, and revised the results and discussions accordingly to strengthen the distinction of the study.

Point 1: The reader understands that there are hospital architectural limitations to physical activity during cancer treatment. The separation into sports and non-sports groups makes sense. On the other hand, in the absence of the questionnaire it is difficult to follow the logic of the work. The internet link does not work, it would be advisable to publish it as an annex.

Response 1:

<Modification and Complementation>

I have fully explained the survey questions in the study method (Line 101-122). Due to the limitations of the paper of journal, I think it is difficult to include a questionnaire. However, I will register the questionnaire written in Korean online.

Thank you

Point 2: The focus of the research needs to be better explained.

Response 2:

<Modification and Complementation>

As requested, the contents have been modified and supplemented.

-  Insertion completed (line 58-69).

- Therefore, based on prior research results that physical activity is effective in curing and recovering cancer, this study was intended to compare and analyze the level of physical activity and the influence of exercise recognition and exercise pharmaceutical according to patients' exercise experience. Through this, we are going to provide important basis data for information delivery to form proper physical activity habits for cancer recovery and healing activities and provide basic data for systematic information provision. It is also intended to prove that the foundation lies in human instinct movement for recovery, health promotion, and maintenance after clinical treatment, and to provide an opportunity to enhance the value of the times for new medical and health services by providing a system that can strengthen them, and to establish a patient-centered health and medical service environment to use it as a basis for public relations and education to pay attention to cancer management at the preventive level as well as recovery.

Thank you

Point 3: Obstruction: do you mean limitation, disruption...

Response 3:

<Modification and Complementation>

As requested, the contents have been modified and supplemented.

- Insertion completed (line 49, 54, 268).

- limitation --> obstruction

- limitation --> obstruction

- limitation --> obstruction

Thank you

Point 4: Recognition: do you mean awareness, knowledge, perception, ability...

Response 4:

<Modification and Complementation>

As requested, the contents have been modified and supplemented.

- Insertion completed (line 60).

- awareness --> recognition

Thank you

Point 5: program obstruction

Response 5:

<Modification and Complementation>

As requested, the contents have been modified and supplemented.

- Insertion completed (line 117).

- Program participation restriction --> program participation obstruction

Thank you

Point 6: hospitalized domestic cancer patients?

Response 6:

<Modification and Complementation>

As requested, the contents have been modified and supplemented.

- Insertion completed (line 125).

- a cancer patient in hospitalization

Thank you

Point 7: Daily life time? What is the concept?

Response 7:

<Modification and Complementation>

As requested, the contents have been modified and supplemented.

- Insertion completed (line 137).

- We divided the levels of physical activity into the following. The hours of physical activity for sedentary life, daily life, and recovery were divided into hours of activities for daily activities, such as washing, going to the bathroom, etc., and hours of physical activity were divided into hours for artificially promoting physical activities for recovery.

Thank you

Point 8: patent probably patient

Response 8:

As requested, the contents have been modified and supplemented.

- Insertion completed (line 265).

- patent --> patient

Thank you

Point 9: diver probably diverse

Response 9:

As requested, the contents have been modified and supplemented.

- Insertion completed (line 267).

- diver --> diverse

Thank you

Point 10: The conclusion should better relate the results to the problematic. The practical implications need to be better described.

Response 10:

As requested, the contents have been modified and supplemented.

- Insertion completed (line 323).

- This study is designed to analyze the obstruction factors that inpatients experience when participating in physical activities, focusing on their personal and environmental characteristics, what are the actual obstacles, and how they relate to the differences between physical activity levels and exercise recognition. Through this, it was attempted to provide the importance of physical activities and environment for the recovery of health and the improvement of quality of life as basic data for the improvement of health arbitration by identifying factors affecting physical activities of inpatients. According to this study, the difference between physical activity levels and exercise recognition and physical activity obstructions based on pre-admission experience of cancer patients was found to be statistically significant in terms of cognitive psychology, social culture, and facility obstructions. Second, the results of hierarchical regression analysis show that the level of physical activity and the influence of exercise recognition on physical activity obstructions based on exercise participation experience showed that sedentary life, intentional movement, and negative exercise recognition were relatively the most important factors of physical activity obstruction in the case of exercise participation experience group, and other positive exercise recognition and the most negative social culture in the case of exercise nonparticipating experience group. That is, in the case of exercise participation experience groups, physical activity restrictions were found to be directly related to physical activity as they were more aware of the habits and effectiveness of exercise acquired through past exercise experiences, and their desire for active physical activity was relatively large. On the other hand, in the case of non-participating groups, it was found to be highly related to physical activity constraints in socio-psychological factors, and how both groups commonly perceive exercise was an important factor in physical activity obstructions. Therefore, hospitals and health care facilities should make efforts to restore the health of patients in specialized cancer nursing hospitals and to extend their lives amid improvement in their quality of life, considering such things as environmental construction to promote physical activities of inpatients, conversion of exercise recognition, support for medical personnel and measures to improve family support, and strategies to enhance patient-centered satisfaction with medical services. Based on the above conclusions, I would like to make the following suggestions.

Thank you

Reviewer 2 Report

This is an interesting study, the presentation of the data and the language, however, can be improved. First of all, the title should be shortened.

Did you age-match patients? Please comment.

A graph depicting main results would be recommended. 

Please discuss the beneficial effects of training on health more extensively: why does training lead to this benefit, e.g. increase in capillary density corresponding to better tissue oxygenation; what are the determinants of exercise capacity?

Further:

In the introduction, please revise sentence 2 (page 1, lines 30-32).

Line 60: EPE and NE should be defined at the first use in the introduction

Unless marking a correction, all letters should be in black (page 2). 

Lines 62/ 63: Based on the survey through the survey (wording), please revise

In summary, this is a valuable study and the manuscript would greatly benefit from language correction by a native speaker.

Author Response

Thank you for your insightful and useful comments.

We specified the research method, corrected the errors in the statistical method, and revised the results and discussions accordingly to strengthen the distinction of the study.

Point 1: This is an interesting study, the presentation of the data and the language, however, can be improved. First of all, the title should be shortened.

Response 1:

<Modification and Complementation>

As requested, the title has been clarified.

- Insertion completed (line 2-3).

- “Obstruction and Recognition of Inpatient Cancer Patients' Physical Activity Based on Exercise Experience”

Thank you

Point 2: Did you age-match patients? Please comment.

Response 2:

<Modification and Complementation>

As you requested, we have added information on the age and number of cases for the research targets.

- Insertion completed (line 75-81)

- “Specific criteria for the selection are adults over the age of 20 who have been diagnosed with cancer and hospitalized in a specialized hospital for treatment. Among them were 8 persons under 40 (EPE 4, NE 4), 32 persons in their 40s (EPE 16, NE 16), 69 persons in their 50s (EPE 34, NE 35), 48 persons in their 60s (EPE 24, NE 24), 37 persons over 70s (EPE 19, NE 18).”

Thank you

Point 3: A graph depicting main results would be recommended.

Response 3:

<Modification and Complementation>

As requested, I graph the main results.

- Insertion completed (line 173-175, 232-237)

Figure 2. Results of Multiple Regression Analysis of Physical Activity Level and Obstruction Factors Based on Physical Activity Participation Experience

Figure 3. Results of hierarchical regression analysis for the Effect of Physical Activity Level and Obstruction Factors Based on Physical Activity Participation Experience(EPE)

Figure 4. Results of hierarchical regression analysis for the Effect of Physical Activity Level and Obstruction Factors Based on Physical Activity Participation Experience(NE)

Point 4: Please discuss the beneficial effects of training on health more extensively: why does training lead to this benefit, e.g. increase in capillary density corresponding to better tissue oxygenation; what are the determinants of exercise capacity?

Response 4:

<Modification and Complementation>

I revised it as you requested.

- Insertion completed (line 308-321)

- It has already been revealed by a number of prior studies that exercise can have a very beneficial effect on cancer patients. Aerobic exercise, in particular, induces the contraction and relaxation of blood vessels while contracting and relaxing muscles. It is said that this enhances blood vessel function, increases blood flow to the heart, strengthens heart function [33], and improves resistance to muscular fatigue by preventing leukocyte reduction [34]. This results in an average maximum oxygen intake of 14 ml/kg/min for cancer patients, which is about 25% of normal people, reaching a level where even a small amount of physical activity can cause severe fatigue [35]. However, after the application of motor therapy, the maximum oxygen intake has increased significantly, which is also positive for the prognosis of patients, and has the effect of improving the musculoskeletal system and mitigating side effects of chemotherapy [36]. It is also effective for the purpose of preventing or treating psychological problems as well as for the recovery of these functions [37].Therefore, in order to help the hospitalized cancer patient recover, the physical activity experience of cancer patients and the mechanism of disability should be carefully analyzed so that appropriate responses to exercise at the health and medical service level can be attempted.

Point 5: In the introduction, please revise sentence 2 (page 1, lines 30-32).

Response 5:

<Modification and Complementation>

I revised it as you requested.

- Insertion completed (line 31)

- korea --> Korea

Thank you

Point 6: EPE and NE should be defined at the first use in the introduction

Response 6:

<Modification and Complementation>

I revised it as you requested.

- Insertion completed (line 61-62).

- exercise participation experience (EPE) group, the non-experience (NE) group

Thank you

Point 7: Unless marking a correction, all letters should be in black (page 2).

Response 7:

<Modification and Complementation>

I revised it as you requested.

Thank you

Point 8: Based on the survey through the survey (wording), please revise

Response 8:

<Modification and Complementation>

Overall, the content has been greatly revised.

- Insertion completed (line 58-69).

- Therefore, based on prior research results that physical activity is effective in curing and recovering cancer, this study was intended to compare and analyze the level of physical activity and the influence of exercise recognition and exercise pharmaceutical according to patients' exercise experience. Through this, we are going to provide important basis data for information delivery to form proper physical activity habits for cancer recovery and healing activities and provide basic data for systematic information provision. It is also intended to prove that the foundation lies in human instinct movement for recovery, health promotion, and maintenance after clinical treatment, and to provide an opportunity to enhance the value of the times for new medical and health services by providing a system that can strengthen them, and to establish a patient-centered health and medical service environment to use it as a basis for public relations and education to pay attention to cancer management at the preventive level as well as recovery.

Thank you

Point 9: In summary, this is a valuable study and the manuscript would greatly benefit from language correction by a native speaker.

Response 9:

<Modification and Complementation>

I agree with your opinion.

After modifying the contents, we are going to request the correction of the MDPI.

Thank you

Reviewer 3 Report

It is a very interesting job but some small changes have to be made to the manuscript.

This article provides relevant information on the characteristics of people with cancer and in relation to physical activity, but there are some aspects that I think should be taken into account:
• The following sentence should be included in the methodology-subjects section:
“Specifically, we divided patients into EPE and NE groups and analyzed the
difference.” (Page 2, line 59-60).

• What kind of criteria was determined to establish the level of physical activity experience?
• It would have been interesting to describe the type of physical activity they were doing.
• The methodology must include that the subjects participated voluntarily and that they were adequately informed of all the research.
• The introductory section should describe the concepts EPE and NE. They have been described in the summary.

Author Response

Thank you for your insightful and useful comments.

We specified the research method, corrected the errors in the statistical method, and revised the results and discussions accordingly to strengthen the distinction of the study.

Point 1: The following sentence should be included in the methodology-subjects section:

“Specifically, we divided patients into EPE and NE groups and analyzed the difference.”

Response 1:

<Modification and Complementation>

I revised it as you requested.

- Insertion completed (line 61).

- exercise participation experience (EPE) group, the non-experience (NE) group

Point 2: What kind of criteria was determined to establish the level of physical activity experience?

Response 2:

<Modification and Complementation>

I revised it as you requested.

- Insertion completed (line 79-81).

- In addition, people who have been exercising for more than three days a week and more than 30 minutes a day for the past year before diagnosing cancer were classified as physical activity experiences.

Thank you

Point 3: It would have been interesting to describe the type of physical activity they were doing.

Response 3:

There were no questions about physical activity types in this survey.

But, we asked only the experience of physical activity participation by Yes and No question.

When we conduct further research, we will add questions about the type of physical activity.

Point 4: The methodology must include that the subjects participated voluntarily and that they were adequately informed of all the research.

Response 4:

I revised it as you requested.

- Insertion completed (line 81-82).

- After explaining the purpose of the study, only those who signed the consent form participated in the study.

Thank you

Point 5: The introductory section should describe the concepts EPE and NE. They have been described in the summary.

Response 5:

I revised it as you requested.

- Insertion completed (line 61).

- exercise participation experience (EPE) group, the non-experience (NE) group

Round 2

Reviewer 1 Report

Thanks to the authors for having greatly facilitated the reading of the text and its comprehension.